# Effect of Silica-Based Nanomaterials on Seed Germination and Seedling Growth of Rice (*Oryza sativa* L.)

**DOI:** 10.3390/nano12234160

**Published:** 2022-11-24

**Authors:** Yaqi Jiang, Jie Yang, Mingshu Li, Yuanbo Li, Pingfan Zhou, Quanlong Wang, Yi Sun, Guikai Zhu, Qibin Wang, Peng Zhang, Yukui Rui, Iseult Lynch

**Affiliations:** 1Beijing Key Laboratory of Farmland Soil Pollution Prevention and Remediation, College of Resources and Environmental Sciences, China Agricultural University, Beijing 100193, China; 2Department of Chemistry, Queen Mary University of London, London E1 4NS, UK; 3School of Geography, Earth and Environmental Sciences, University of Birmingham, Edgbaston, Birmingham B15 2TT, UK; 4China Agricultural University Professor’s Workstation of Yuhuangmiao Town, Shanghe County, Jinan 250061, China; 5China Agricultural University Professor’s Workstation of Sunji Town, Shanghe County, Jinan 250061, China

**Keywords:** silicon-based nanomaterials, rice, plant growth, seed germination

## Abstract

The application of nanomaterials (NMs) in agriculture has become a global concern in recent years. However, studies on their effects on plants are still limited. Here, we conducted a seed germination experiment for 5 days and a hydroponics experiment for 14 days to study the effects of silicon dioxide NMs(nSiO_2_) and silicon carbide NMs(nSiC) (0,10, 50, 200 mg/L) on rice (*Oryza sativa* L.). Bulk SiO_2_ (bSiO_2_) and sodium silicate (Na_2_SiO_3_) were used as controls. The results showed that nSiO_2_ and nSiC increased the shoot length (11–37%, 6–25%) and root length (17–87%, 59–207%) of germinating seeds, respectively, compared with the control. Similarly, inter-root exposure to nSiO_2_, bSiO_2,_ and nSiC improved the activity of aboveground catalase (10–55%, 31–34%, and 13–51%) and increased the content of trace elements magnesium, copper, and zinc, thus promoting the photosynthesis of rice. However, Na_2_SiO_3_ at a concentration of 200 mg/L reduced the aboveground and root biomass of rice by 27–51% and 4–17%, respectively. This may be because excess silicon not only inhibited the activity of root antioxidant enzymes but also disrupted the balance of mineral elements. This finding provides a new basis for the effect of silica-based NMs promotion on seed germination and rice growth.

## 1. Introduction

As the global population continues to grow, the demand for rice is gradually increasing and the world is overwhelmed with the demand for rice [1]. Under such harsh circumstances, it is urgent to improve the rice yield per unit of land [2]. Although scientists have made their best efforts to conduct in-depth research, they have not been able to increase production successfully. One of the main reasons for the low yield is the low efficiency of fertilizer use. How to improve the utilization rate of existing fertilizer has become a major problem that must be solved to increase the yield. In recent years, nanotechnology has appeared in our daily life, including in medicine, aerospace, industrial materials, and so on [3]. Notably, nanotechnology can significantly transform food production in agriculture around the world [4]. Nanomaterials (NMs) refer to the material in which at least one of the three dimensions is at the nanoscale, that is 1–100 nm. Due to their nanoscale dimensions, NMs have high surface-to-volume ratios and thus very specific properties [5]. Previous studies have shown that NMs can also be used as fertilizer, because they can release a small number of metal ions to supplement nutrition for plants or improve plant tolerance to promote plant growth [6,7]. In addition, nanofertilizers prevent nutrient leaching or volatilization, thereby maintaining soil fertility longer than conventional fertilizers [8].

Silicon (Si) is the second richest element in the earth’s crust [9]. Si exists mainly in the form of mineral silicates, aluminosilicates, and silicon dioxide (SiO_2_). However, most of these forms cannot be absorbed by plants. They can only be absorbed in the form of monosilicate (H_4_SiO_4_) [10]. Si is generally considered a beneficial element that aids plant growth and relieves various abiotic and biotic stresses [11]. Rice can absorb Si up to 10% of its dry weight, which is significantly more than the uptake of the macronutrients such as nitrogen, phosphorus, and potassium [12]. Therefore, to achieve high yields in rice, additional Si needs to be added artificially. The deficiency of Si in rice can reduce yield and predispose it to the lodge. In addition, chronic consumption of Si-deficient foods can adversely affect human health [13]. Although the effect of Si on plants is an emerging area of research, there are not enough studies comparing the uptake and transport of Si-based NMs and conventional Si fertilizers in plants, and their effects on plants are controversial. Previous research showed that Si-based NMs were found to be toxic only at very high concentrations. Lin et al. [14] found that SiO_2_ NMs at 500 μL/L also increased the biomass of forest trees and promoted root growth in 2004. In addition, Lee et al. [15] found that Si NPs promoted *Arabidopsis* root elongation at low concentrations (400 mg/L), but at higher concentrations (>2000 mg/L) resulting in toxicity. Since we wanted to explore the possibility of Si-based NMs as nanofertilizers, we chose concentration intervals with lower concentrations of elemental Si, 0,10,50 and 200 mg/L, respectively. In this study, we used rice as the research target and hypothesized that Si-based NMs (SiO_2_ NMs and SiC NMs) could be used as a more effective and better Si-based fertilizer than the conventional Si-based fertilizer (sodium silicate and bulk SiO_2_) to promote rice seed germination and seedling growth. Previous studies showed that Si-NPs increased the photosynthetic pigment content of black wheat under heat stress [16]. The content of chlorophyll is related to the photosynthesis of plants, which affects the growth and development of rice to some extent [17]. Therefore, in this paper, the relative chlorophyll content changes of rice leaves treated with different Si-based materials were explored by measuring SPAD value. On this basis, the content changes of chlorophyll of different species (chlorophyll a, chlorophyll b, and total chlorophyll) were determined, respectively. In addition, we hypothesized that Si-based NMs can release Si ions slowly and sustainably, rather than releasing toxic Si ions at excessively high concentrations. We analyzed the antioxidant system and mineral element content to explore the molecular mechanisms of the observed effects. There is still no knowledge about the effect of SiC NMs on rice growth and development. To our knowledge, this is the first study to demonstrate that Si-based NMs act as nanofertilizers to enhance rice seed germination and seedling growth.

## 2. Materials and Methods

### 2.1. Characterization of Si-Based NMs

SiO_2_ NMs and SiC NMs (20–30 nm, purity >99.9%) were purchased from Pantian NMs Co., Ltd. (Shanghai, China). All other chemical reagents in the experiment were purchased from Beijing Chemical Factory (Beijing, China). The morphology and size of the two Si-based NMs were characterized by transmission electron microscopy (TEM, JEOL, Tokyo, Japan) and scanning electron microscopy (SEM, AJSM-F100, Tokyo, Japan). The hydrodynamic sizes and zeta potential in deionized water were characterized by Malvern Zetasizer (Nano ZS90, Malvern, UK) and sonicated for 30 min before characterization. Fourier transform infrared (FT-IR; Nicolet iS20, Thermo Fisher Scientific Inc., USA) analysis was finished at a range of 400–4000 cm^−1^.

The average sizes of SiO_2_ NMs and SiC NMs were 20.28 ± 4.5 nm and 37.9 ± 8.9 nm, respectively. The hydrodynamic sizes of SiO_2_ NMs and SiC NMs were 495.9 ± 19.5 nm and 2487 ± 30 nm, and the zeta potentials were −28.03 ± 2.83 mv and −24.96 ± 0.7 mV, respectively. As shown in Appendix A, SiO_2_ NMs have a smaller size compared with SiC NMs, which implies a larger specific surface area and adsorption capacity. In addition, the zeta potential results of SiO_2_ NMs indicated that SiO_2_ NMs are more capable of adsorbing metal cations than SiC NMs.The peak of SiO_2_ at 3431 cm^−1^ is the tensile vibration absorption peak of the Si-O-H bond, the bending vibration peak of -OH at 1638 cm^−1^, the asymmetric tensile vibration absorption peak of Si-O bond at 1101 cm^−1^, and the bending vibration peak of Si-O-Si bond at 470 cm^−1^. Appendix A showed that there is a strong absorption peak at 830 cm^−1^. This is the characteristic absorption peak of SiC corresponding to the stretching vibration of Si-C bond.

### 2.2. Seed Germination

Seeds of hybrid rice (Y Liangyou 900) were purchased from the Chinese Academy of Agricultural Sciences (Beijing, China). Y Liangyou 900 is an indica-type two-series hybrid rice with National Trial No. 2015034. Y Liangyou 900 is a fourth-stage super hybrid rice combination produced by combining Y58S (the backbone sterile line of super hybrid rice) and R900 (a light-sensitive and strongly superior recovery line selected from the indica-japonica cross). Furthermore, Y900 is widely used in agricultural production for its high resistance to overturning and record yield. Rice (Y Liangyou 900) seeds of uniform size were selected, soaked in 10% hydrogen peroxide(H_2_O_2_) solution for 30 min, and then rinsed with deionized water to ensure surface sterility. Aqueous suspensions of SiO_2_ NMs(nSiO_2_), bulk SiO_2_(bSiO_2_), sodium silicate (Na_2_SiO_3_), SiC NMs(nSiC) were prepared at concentrations of 10, 50, and 200 mg/L and dispersed in an ultrasonic bath for 30 min before exposure. A broad concentration range was chosen to ensure that relevant phytotoxic responses were observed at high doses and possible positive effects at low doses. A sheet of filter paper was placed in a 9 cm × 2 cm Petri dish and immersed in 5 mL of Si-based material suspension or deionized water as a control [18]. Sixteen seeds were arranged in each Petri dish with even spacing between seeds. All Petri dishes were sealed tightly with film to prevent water loss during germination. In the climate chamber, seeds were allowed to germinate in the dark at 27 °C. For each treatment, four replicates were germinated for 5 days. There were 4 replicates for each treatment. To calculate the germination rate, the length of roots and shoots was measured with a meter ruler. 

### 2.3. Plant Growth and Treatment

The seeds (Y Liangyou 900) were sterilized in 10% H_2_O_2_ solution for half an hour, rinsed with distilled water, and then placed on a gauze-lined tray and poured with an appropriate amount of distilled water to keep the filter paper moist. Then, placed in a constant temperature incubator at 27 °C and 70% humidity for 48 h in dark incubation until the seeds appeared white. Then, incubated at 25 °C and 75% humidity for 4 days. Five well-grown and uniform rice seedlings were fixed on sponges and transferred to a 250 mL bottle containing 1/2 strength Kimura B nutrient solution (pH = 5.5). The seedlings were allowed to grow for 7 days in a 16 h light (27 °C and 70% humidity) and 8 h dark (25 °C and 70% humidity) cycle. The nutrient solution was renewed daily. After 7 days, seedlings were treated with SiO_2_ NMs (nSiO_2_), bulk SiO_2_(bSiO_2_), Na_2_SiO_3_, and SiC NMs(nSiC) with Si concentrations of 10, 50, and 200 mg/L. Before treatment, the NMs suspension was sonicated for 30 min to disperse. Seedlings treated with only Kimura B nutrient solution were set as the control group. The solution in the bottle was replenished once a day with a freshly prepared ½ kimura solution and mixed. Seedlings were allowed to grow for another 10 days before harvest. There were 4 replicates for each treatment. Leaves and roots were thoroughly rinsed with 0.1% HNO_3_ and deionized water to remove adsorbed NMs. Length and fresh biomass of shoots and roots were measured immediately with a 1/10,000 scale and meter ruler after harvest. Dry weights were measured after drying in an oven (1 h at 105 °C and 48 h at 65 °C).

### 2.4. Chlorophyll Content

The first fully expanded leaf in rice was selected and eight points near the main vein were measured using a chlorophyll meter (Konic Minolta, Japan). SPAD value was measured to indicate relative chlorophyll content in leaves. A total of 40 values were averaged for each group of five seedlings. Chlorophyll content was measured using the assay kit purchased from Nanjing Jiancheng Co (China). First, 0.1 g of fresh rice leaves were cut into shreds. Then, the chlorophyll fraction was extracted into 5 mL of anhydrous ethanol and acetone (1:2 *v*:*v*) solution and immersed in the dark for 3 h. The absorbance was measured at 663 nm and 645 nm. Chlorophyll a (Chla), chlorophyll b (Chlb), and total chlorophyll content (mg/g) were calculated using the following equations [19]:(1)$$C_{\mathrm{Chla}}=\left(12.7A_{663}-\;2.69A_{645}\right){\;\times \;V}_{\mathrm{Extraction}\;\mathrm{solution}}\;\times \;\frac{N}{m}/1000$$
(2)$$C_{\mathrm{Chlb}}=\left(22.9A_{645}-4.68A_{663}\right){\;\times \;V}_{\mathrm{Extraction}\;\mathrm{solution}}\;\times \;\frac{N}{m}/1000$$
(3)$$C_{\mathrm{Total}\;\mathrm{Chlorophyll}}=\left(20.21A_{645}+8.02A_{663}\right){\;\times \;V}_{\mathrm{Extraction}\;\mathrm{solution}}\;\times \;\frac{N}{m}/1000$$

Note: C_Chla_: The content of Chlorophyll a (mg/g); C_Chlb_: The content of Chlorophyll b (mg/g); C_Total Chlorophyll_: The content of total chlorophyll (mg/g); A_663_: The absorbance was measured at 663 nm; A_645_: The absorbance was measured at 645 nm; V_Extraction solution_: The volume of the extraction solution; N: Dilution times; m: Sample mass (g).

### 2.5. Antioxidant System

The activities of antioxidant enzymes including peroxidase (POD), superoxide dismutase (SOD), and catalase (CAT) were examined using the assay kits purchased from Nanjing Jiancheng Ltd. (Nanjing, China). Malondialdehyde (MDA) content was also measured, as described by Jiang et al. [20]. 

#### 2.5.1. POD Activity

A mixture containing 0.1 mL of supernatant, 2.4 mL of reagent A, 0.3 mL of reagent B, and 0.2 mL of reagent C was placed in a 37 °C water bath for 30 min. Then, 1.0 mL of reagent D was added. After centrifugation at 3500 rpm for 10 min, the supernatant was left to stand, and the absorbance was measured at 420 nm.

#### 2.5.2. SOD Activity

Added 20 μL of xanthine oxidase working solution and 200 μL of water-soluble tetrazolium salt solution to 20 μL of supernatant. The mixed samples were incubated at 37 °C for 40 min, and then the absorbance was measured at 530 nm.

#### 2.5.3. CAT Activity

A total of 20 μL of enzyme extraction was mixed with 2.5 mL of the CAT reaction solution. Then, the absorption at 240 nm was recorded immediately and after 1 min. 

#### 2.5.4. MDA Content

Mixed 0.2 mL of supernatant, 0.2 mL of reagent A, 3.0 mL of reagent B, and 1.0 mL of reagent C in 95 °C water for 40 min. After cooling to room temperature, the samples were centrifuged at 4000 rpm for 10 min. The absorbance of the supernatant was measured at 532 nm.

### 2.6. Mineral Element Content Analysis

Mineral elements were measured, as described by Rui et al. [21]. Briefly, rice plants with different treatments were harvested, thoroughly washed, and dried at 65 °C to prepare 0.2 g dry rice samples (roots and aboveground) for the determination of mineral elements. Dried rice samples were ground into powder and thoroughly digested in a mixed solution of nitric acid: hydrofluoric acid (1:2) using a microwave digestion system (XT-9916, XTrust, Xiamen, China). The digests were diluted to 10 mL with ultrapure water and assayed by ICP-MS (DRCII, PerkinElmer, and Norwalk, CT, USA).

### 2.7. Statistical Analysis

Data were analyzed using SPSS 25.0 software for one-way analysis of variance (ANOVA). *p* ≤ 0.05 was regarded statistically significant using Duncan’s test for comparison of means.

## 3. Results and Discussion

### 3.1. Effects of Si-Based NMs on Seed Germination

As shown in Figure 1, none of the Si-based materials had a significant effect on the seed germination rate. The concentration of 10 mg/L nSiO_2_ significantly increased shoot elongation, while Na_2_SiO_3_ and nSiC treatments inhibited shoot elongation. nSiO_2_ at 50 mg/L and 200 mg/L still significantly increased the root length by 69% and 87%, respectively. Notably, 50 mg/L and 200 mg/L nSiC also increased root length by 207% and 90%. In addition, the root length was also significantly increased by the bSiO_2_ treatment, reaching a maximum concentration of 50 mg/L. However, root elongation was inhibited by 200 mg/L Na_2_SiO_3_.

Thus, it can be concluded that Si-based materials affect seed germination differently at different concentrations. The 10–200 mg/L nSiO_2_ favored the germination of rice seeds without adverse effects, probably because NMs increased the water permeability of seed coat or cell walls or membranes of cell organelles [22]. However, the Na_2_SiO_3_ treatment at 200 mg/L produced a series of adverse effects on seed germination. This is because Na_2_SiO_3_ provided a large amount of Si ions in a short period creating an ionic competition that affected the uptake of other elements by the plant. This is consistent with Zhang et al. that high concentrations of NMs lead to excessive ion leakage and thus inhibit seed germination [18]. The different results of seed shoot length and root length under different treatments may be due to the fact that germinating seed shoots grow upward, so NMs have less effect on shoot growth at a later stage. Additionally, the roots continued to be immersed in NMs, so the roots could continue to absorb NMs and achieve better growth. 

However, not all kinds of NMs can promote seed germination. For example, Lin and Xing [23] found that nano-ZnO and nano-Zn at 20 and 200 mg/L did not promote seed germination and seedling root growth, but showed significant inhibition at 2000 mg/L. Previous studies showed that even some NMs inhibit seed germination. Zhang et al. [24] investigated the effects of six biochar nanoparticles (BNPs) on seed germination and growth of rice, tomato, and reed seedlings. It was found that BNPs collected from high temperature (700 °C) biochar inhibited seed germination of rice. In addition, it significantly reduced branch length and biomass.

### 3.2. Effects of Si-Based NMs on the Growth of Rice Seedlings

After germination trials, we further explored the effects of inter-root exposure to Si-based NMs on rice seedlings. The aboveground and roots of rice seedlings treated with Si-based NMs showed different growth trends.

Compared with the control, 200 mg/L nSiC treatment maximized the growth of rice seedlings, increasing shoot height and aboveground biomass by 22% and 68%, respectively (Figure 2). While 200 mg/L Na_2_SiO_3_ significantly decreased the shoot height and aboveground biomass by 24% and 36%. The growth-promoting effect of nSiC on rice comes from the release of Si on the one hand and may come from the transfer of small-sized nSiC within the rice on the other hand. While the inhibitory effect of Na_2_SiO_3_ may be mainly due to the release of excessive ionic Si in a short period causing some oxidative damage to rice. In addition, the shoot height of rice seedlings showed a dose-dependent increase treated with nSiO_2_. A total of 200 mg/L of nSiO_2_ increased shoot height by 7%, and the aboveground biomass showed the same increasing trend as the shoot height. The rice seedlings treated with bSiO_2_ were not significantly different from the control group.

For roots, nSiO_2_ treatments of 50 mg/L and 200 mg/L significantly increased root biomass by 37% and 47% but had no significant effect on root length. Similarly, nSiC treatment at 200 mg/L also significantly increased root length and root biomass. In contrast, bSiO_2_ and Na_2_SiO_3_ treatments did not promote root growth. This may be due to the small particle size and large specific surface area of the NMs, allowing part of the NMs to enter the roots directly to promote root growth, or to adsorb other beneficial elements from the nutrient solution to reach the roots together [25]. However, previous studies have shown that not all NMs promote rice growth. Li et al. [26] compared the physiological impacts of three different Fe-based NPs formulations: zero-valent iron (ZVI), Fe_3_O_4,_ and Fe_2_O_3_ NPs, on hydroponic rice after root exposure for 2 weeks. The results showed that Fe-based NPs treatment did not affect root length, surface area, and root diameter. Furthermore, Thuesomba et al. [27] found that cell damage was evident when rice seeds were exposed to high concentrations of Ag NPs at 100 mg/L. Additionally, root exposure to 100 mg/L Ag NPs was significantly phytotoxic to rice seedlings.

### 3.3. Effects of Si-Based NMs on Chlorophyll Content in Rice Seedlings

SPAD values can be used to characterize the relative chlorophyll levels of leaves and are highly correlated with leaf photosynthetic traits [28]. However, the relative chlorophyll content cannot be completely equivalent to the chlorophyll content, nor can represent the change amount and trend of chlorophyll a and chlorophyll b. Therefore, we then determined the chlorophyll a, b, and total chlorophyll content of the leaves to further explain the photosynthesis and crop growth of the leaves after Si-based NMs treatment [29].

Except for the Na_2_SiO_3_ treatment, the SPAD values of rice leaves increased to some extent with the treatment of Si-based materials and then showed a regular decrease with increasing concentration. The nSiO_2_ and bSiO_2_ at 10 mg/L and 50 mg/L increased the SPAD by 14–25%, and 8–28%, respectively (Figure 3). It is known that a higher SPAD value led to higher rates of photosynthesis. Grains get a higher supply of carbohydrates through photosynthesis, and at the same time, more carbohydrates are sent to the roots to maintain their metabolism [30]. This may also be the reason for the increase in aboveground plant height and biomass of rice under nSiO_2_ and bSiO_2_ at 10 mg/L and 50 mg/L treatments. However, 200 mg/L Na_2_SiO_3_ significantly decreased the SPAD values. The same result was observed for chlorophyll content, which decreased with increasing concentration of Si-based materials. The greatest decrease in photosynthetic pigments was observed at 200 mg/L Na_2_SiO_3_ treatment. Plants produce reactive oxygen species (ROS) during photosynthesis [31], so antioxidant systems are needed to eliminate this stress response in plants. The reduction in photosynthetic pigments by high concentrations of Si-based materials is most likely due to the series of oxidative stresses caused in rice. 

### 3.4. Effects of Si-Based NMs on the Antioxidant System of Rice Seedlings

The antioxidant system is a self-defense system that protects cells when plants are stimulated by external sources. It ensures efficient ROS detoxification, reduces lipid peroxidation in membranes, and prevents protein damage by delaying oxidation and controlling DNA and nucleic acid damage under stress [32]. Therefore, we explored the activities of antioxidant enzymes to respond to stress and tolerance to stress in rice plants after treatment with various Si-based materials.

The aboveground and underground enzyme activities showed different trends, which were consistent with the study of Zhou et al. [33]. This may be because, in the form of rhizosphere exposure, the concentration of NMs is too high in the root, resulting in more drastic changes in root enzyme activity. In the aboveground parts, the CAT activity significantly increased by 28% and 55% under the treatment of 50 and 200 mg/L nSiO_2_. Similarly, the CAT activity also increased by 51% under 50 mg/L nSiC treatment. CAT scavenges plant mitochondrial electron transport, β-oxidation of fatty acids, and H_2_O_2_ produced during photorespiratory oxidation. Increasing evidence suggests that CAT has a vital function in plant defense, senescence, and aging [34]. It showed that a certain number of Si-based NMs were transported to the shoot through the root system, which had a certain stimulating effect on the shoot and increased the activity of CAT. Previous studies have shown that TiO_2_ NPs increased CAT activity in plants and then can protect chloroplast membranes from ROS damage [35]. The increase in the aboveground biomass of rice under low concentrations of nSiO_2_ and nSiC treatment verified that Si-based NMs did not produce oxidative stress that was detrimental to plant growth. Neither nSiO_2_, bSiO_2_ nor nSiC rhizosphere exposure resulted in significant changes in SOD activity. SOD is considered to be the first line of defense against oxidative stress in plants and plays the most important role in scavenging ROS generated under stressful conditions. It does this by catalyzing its decomposition to H_2_O_2_ and finally to H_2_O and O_2_ [34]. However, 10 mg/L Na_2_SiO_3_ significantly increased the activity of SOD, and when the concentration continued to increase to 200 mg/L, the activity decreased significantly by 56–61%. This may be due to the high solubility of Na_2_SiO_3_ caused oxidative stress which inhibited the activity of the enzyme.

For roots, almost all treatments increased the activity of CAT. Notably, the concentration of 50 mg/L nSiO_2_ increased the activity of CAT the most (259%). The changes in POD and SOD activities were the same as CAT, except for the inhibitory effect of the concentration of 200 mg/L Na_2_SiO_3_ on SOD activity. The function of POD is to break down H_2_O_2_. The enzyme activity decreased in a concentration-dependent manner under the treatment of bSiO_2_ and Na_2_SiO_3_ but was still higher than that of the control group. Similar results were obtained for rice inter-root exposed to different concentrations of CuO NMs [31]. The results showed that CuO NMs above 50 mg/L caused oxidative damage to rice plants, while lower concentrations had no significant effect on the physiological and biochemical traits of rice. The enzyme activity of nSiO_2_ was maintained at a high level at all concentrations, which indicated that the high concentration of nSiO_2_ did not negatively affect the enzyme activity. MDA is an important indicator of cell membrane damage [36]. It can be seen from Figure 4, the trends of MDA content and the activities of SOD and POD in rice plants treated with different Si-based materials were very similar. This indicated that the treatment of Si-based materials produced some oxidative stress on rice, which increased the MDA content. However, there was no significant adverse effect of antioxidant enzymes on plant growth under Si-based NMs treatment, indicating that NMs modulated the antioxidant system and gradually alleviated the oxidative stress of rice.

### 3.5. Effects of Si-Based NMs on Mineral Element Content in Rice Seedling

Macronutrient and micronutrient contents of rice aboveground and roots were measured during all treatments. This is because various micronutrients are beneficial elements for plant growth and development and can be involved in various processes of plant growth and development, especially photosynthesis [37]. Therefore, we determined the mineral element content to show the effect of Si-based NMs on rice. A total of 50 mg/L of nSiO_2_ and bSiO_2_ significantly increased the magnesium (Mg) content aboveground. While the content started to decrease as the concentration increased to 200 mg/L. In addition, a series of changes in the contents of copper (Cu), iron (Fe), and zinc (Zn) in rice were observed under different concentrations of Si-based materials (Figure 5).

Treatment with 50 mg/L of nSiO_2_ and bSiO_2_ significantly increased the content of Mg by 6% and 11% in aboveground parts. Interestingly, the chlorophyll content in the roots was also increased under the treatment of 50 mg/L of nSiO_2_ and bSiO_2_. This is because Mg plays a key role in photosynthesis since it is an important component of chlorophyll, a light-absorbing green pigment found in plants [38]. However, in the roots, the uptake of Mg increased significantly only with 10 mg/L of nSiO_2_ and bSiO_2_ treatment, the other treatment showed an inhibitory trend.

Cu is essential for photosynthetic processes, critical for plant respiration, and can assist in the metabolism of carbohydrates and proteins [39]. The nSiO_2_ and bSiO_2_ at 10 mg/L and 50 mg/L caused a dose-dependent increase in the Cu content on the ground, and the upward trend decreased when the concentration reached 200 mg/L. Na_2_SiO_3_ treatment severely inhibited Cu uptake in the aboveground parts of rice, which led to a decrease in the activity of related enzymes and photosynthesis. In contrast, the uptake of Cu by rice increased by 35% and 23% when the roots were exposed to 10 mg/L of nSiO_2_ and nSiC, respectively. The increase in Cu content in rice roots exposed to 200 mg/L Na_2_SiO_3_ may be due to the disruption of the cell membrane structure by the excessive Si ion concentration, allowing Cu ions from the nutrient solution to enter the roots directly. However, the increased Cu content in rice roots exposed to 200 mg/L nSiO_2_ may be due to the large specific surface area of NMs, which adsorbed Cu from the solution into the roots [25]. Additionally, in the nSiO_2_ treatment group, the content of Cu was better because the zeta potential of nSiO_2_ was greater than that of nSiC and the adsorption force was stronger.

The content of Zn in the shoots of rice seedlings treated with nSiO_2_ and bSiO_2_ both increased and reached its maximum at a concentration of 50 mg/L, increasing by 136% and 42%, respectively. Zn plays an important role in the metabolism of plants and its deficiency can cause many adverse effects on the growth of plants [40]. It has been shown that Zn is valuable as a yield component in wheat and can help improve the water use efficiency of plants if applied in moderation [41]. A low concentration of NMs treatment promoted the accumulation of Zn, thus enhancing plant growth and development. However, the uptake of Zn in the roots was reduced under all treatments.

Fe is an essential element in plant cellular respiration, metabolism, DNA synthesis, and photosynthesis [42]. Fe deficiency can seriously affect human health and lead to anemia. Fe is the most deficient trace element in the human diet and approximately 2 billion people in the world are affected by Fe deficiency [43]. At 50 mg/L nSiO_2_ treatment, Fe content in rice seedling shoots treated with bSiO_2_ significantly increased by 11% and 10%, while Na_2_SiO_3_ and nSiC decreased Fe content. The chlorophyll content also decreased under this treatment. In photosynthesis, Fe is a key factor in carbon dioxide fixation, and Fe deficiency can lead to a decrease in SPAD values, which can lead to the yellowing of leaves. This is why the rice leaves treated with 200 mg/L Na_2_SiO_3_ appear yellow (Appendix A). Low concentrations of NMs can promote Fe uptake and improve photosynthesis. However, when the concentration is too high, too much ROS is produced inside the plant, causing micro-toxicity that is detrimental to plant growth and development. This is consistent with the research of Motyka et al. that the addition of nano-titanium dioxide may block the root system and reduce the chlorophyll content in mint leaves [44]. The rhizosphere exposure conditions resulted in more Si-based materials in the roots, and greater root stress, only nSiO_2_ treatment of 10 mg/L and 50 mg/L increased the Fe content. This also confirmed that nSiO_2_ enhanced the stress resistance of rice to a certain extent. Similarly, El-Shetehy et al. demonstrated that SiO_2_ NMs can induce systemic acquired resistance in a dose-dependent manner, which involved the defense hormone salicylic acid and enhanced stress resistance [45].

Manganese (Mn) plays a crucial role in photosynthesis by helping chlorophyll synthesis and is also important in maintaining enzyme activity. The Mn content aboveground showed a largely suppressive trend. In the roots, 10 mg/L of nSiO_2_, and bSiO_2_ promoted the uptake of Mn in rice, which was also verified by the enzymatic activity of the antioxidant system. In addition, Mn affects the uptake and assimilation of other nutrients by plants. This is consistent with the results of Faraz et al. [46], where a low concentration of NMs treatment promoted the uptake of Mn in the roots and thus also increased the enzyme activity in the roots.

The content of Se in the shoots was lower than that in the control group, but in the roots, the content of Se was significantly increased in the nSiO_2_, bSiO_2,_ and nSiC groups. Increased selenium (Se) levels in plants may suppress nitrogen, phosphorus, and sulfur concentrations in tissues. It also inhibits the absorption of several heavy metals, especially Mg, Zn, Cu, Fe, and cadmium [47]. This may be one of the reasons for the reduced content of other mineral elements in the roots.

## 4. Conclusions

This paper showed that the Si-based NMs had little effect on seed germination rate but could increase the sprout and root length of germinated seeds. In hydroponics experiments, low concentrations of Si-based NMs can eliminate ROS by increasing antioxidant enzyme activity, thus contributing to the growth of rice plants. However, too high a concentration of Si ions can destroy the internal structure of rice and inhibit the accumulation of photosynthetic pigments and the absorption of trace elements. In this study, SiO_2_ NMs and SiC NMs were considered for the first time in comparison with conventional Si fertilizers to propose the idea of Si-based NMs as nanofertilizers. This finding provides new insights and a basis for the safe application of Si-based NMs in agriculture. Future research needs to integrate physiological and biochemical effects using molecular techniques to further explore the specific mechanisms by which Si-based NMs affect seed germination and seedling growth, as well as the feasibility of Si-based NMs as nanofertilizers.

## Figures and Tables

**Figure 1 nanomaterials-12-04160-f001:**
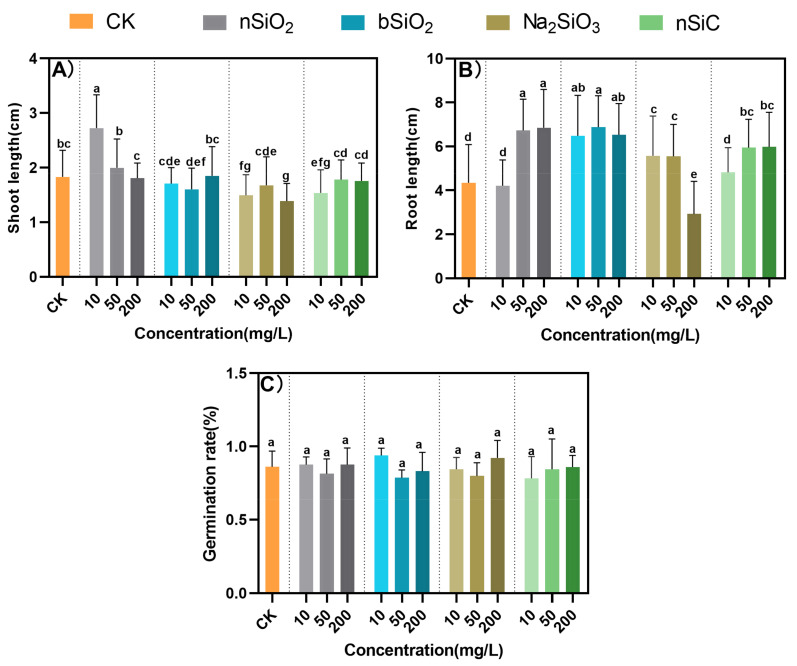
Effects of Si-based NMs on rice seed germination. (**A**) Seed sprout length; (**B**) Seed root length; (**C**) Seed germination rate. Columns marked with the same letter were not significantly different at *p* < 0.05 (*n* = 4).

**Figure 2 nanomaterials-12-04160-f002:**
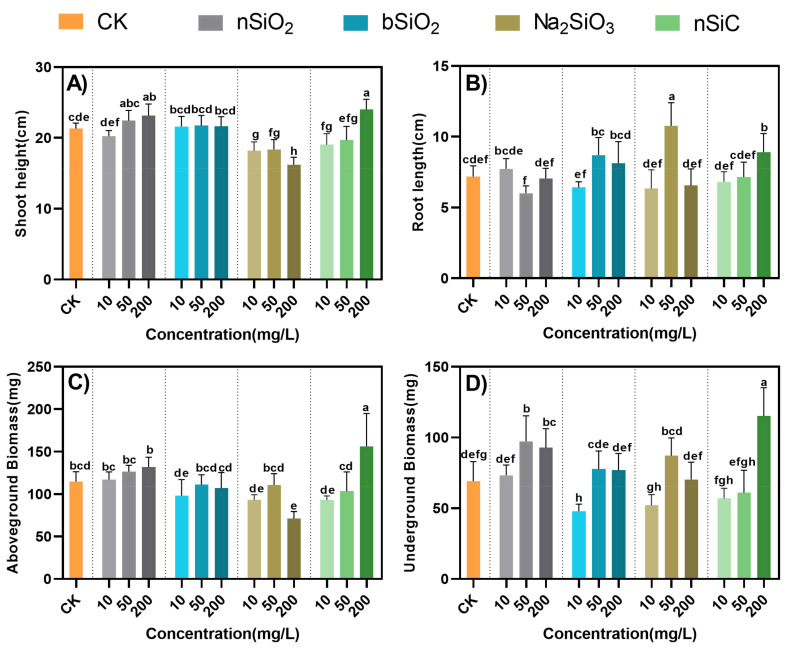
Effects of Si-based NMs on rice growth and development. (**A**) Shoot height; (**B**) Root length; (**C**) Aboveground biomass; (**D**) Underground biomass. Columns marked with the same letter were not significantly different at *p* < 0.05 (*n* = 4).

**Figure 3 nanomaterials-12-04160-f003:**
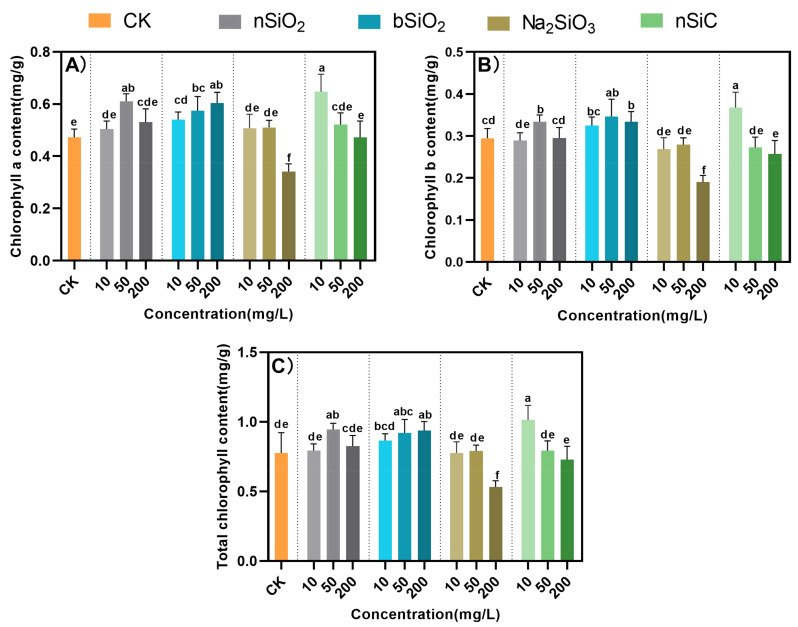
Effects of Si-based NMs on photosynthesis in rice leaves. (**A**) Chlorophyll a; (**B**) Chlorophyll b; (**C**) Total chlorophyll. Columns marked with the same letter were not significantly different at *p* < 0.05 (*n* = 4).

**Figure 4 nanomaterials-12-04160-f004:**
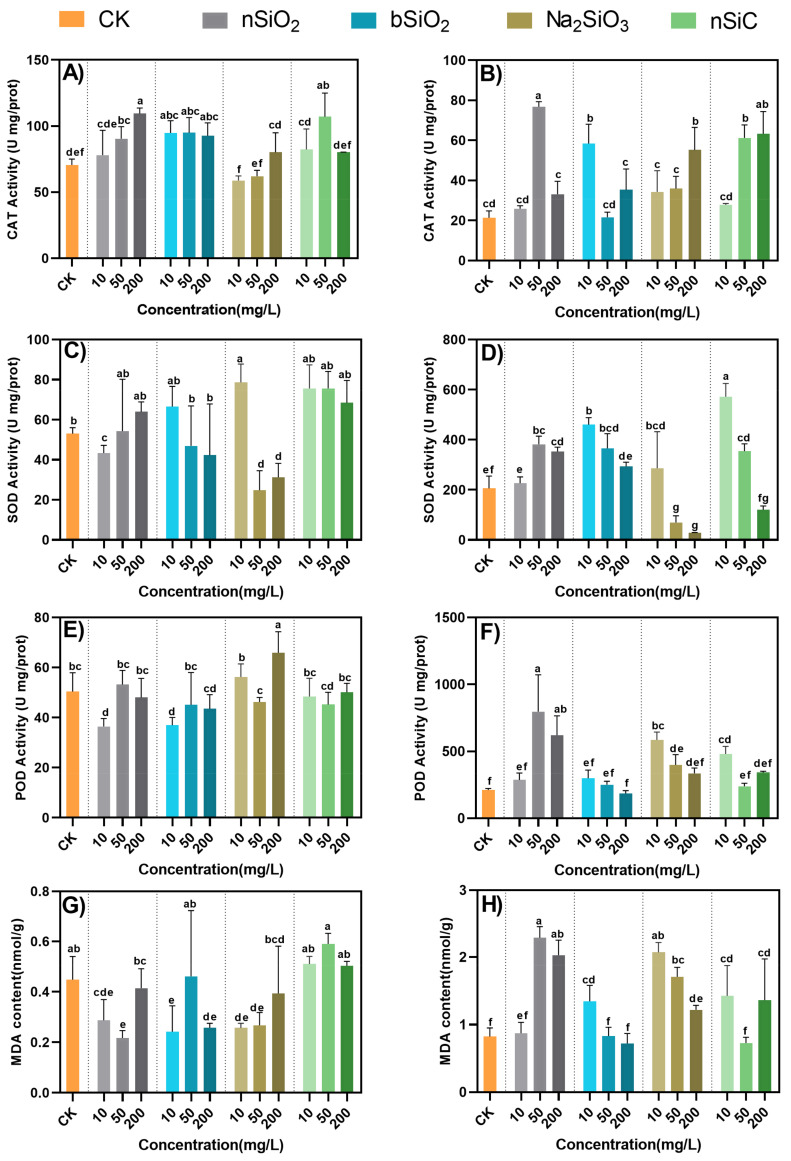
Effects of Si-based NMs on the antioxidant system of rice. CAT, SOD, POD activities and MDA content (**A**,**C**,**E**,**G**) of leaves; (**B**,**D**,**F**,**H**) of roots. Columns marked with the same letter were not significantly different at *p* < 0.05 (*n* = 4).

**Figure 5 nanomaterials-12-04160-f005:**
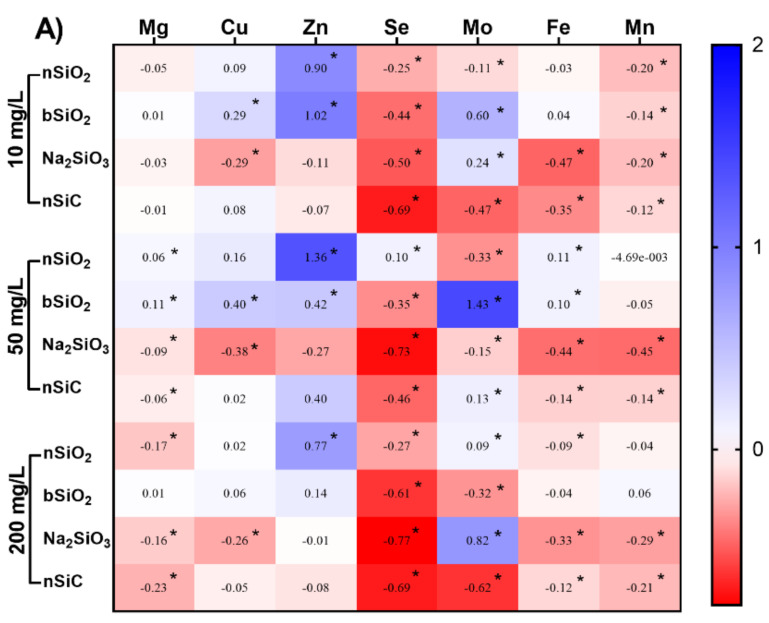
The heat map reflects the percentage increase or decrease of mineral element content in Si-based NMs treated rice (the shoot is (**A**) and root is (**B**)) compared to the control (CK) group. * Indicates significant difference compared to CK group at *p* < 0.05.

## Data Availability

The authors declare that all data supporting the findings of this study are available within the article (and its Appendix A).

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
