# Peer review of "Effect of Silica-Based Nanomaterials on Seed Germination and Seedling Growth of Rice (Oryza sativa L.)"

_nanomaterials, 2022, doi:10.3390/nano12234160_

Round 1

Reviewer 1 Report

In your paper “Effect of silica-based nanomaterials on seed germination and seedling growth of rice (Oryza sativa L.)” you report results from 2 experiments focussing on the effects of Si applied either as nanomaterial or as “normal” bulk material.

The topic of this manuscript is clearly within the scope of Nanomaterials. However, the length of the paper is not is an adequate relation to the scientific novelty. Although still rather few studies are available focussing on nanomaterials in the context of plant nutrition, the approach and the way you present your results is not fully convincing.

Overall the structure of the paper is acceptable and the title of the paper is clearly understandable.

The abstract is not written in a way that readers get a clear picture on what you have done in your experiments and they are not encouraged to read the full article. To my understanding you must give more details on the experimental setup (2 different lab trials, parameter measured). Furthermore, most reader would appreciate to get the facts presented in a more understandable way. The final concluding sentence is too unspecific!

The keywords are not all adequate (do NOT repeat terms already used for the title; do not use non-specific terms like “agriculture” or “mineral elements”).

The Introduction is okay, BUT at the end of this section, you MUST clearly state your objections and add 2 – 3 hypotheses! For the reader this is very important because based on the hypotheses you can enable the reader to understand why you have measured those parameters you describe in the M&M section (e.g. for me it is not clear why you have measured SPAD values AND Chlorophyll concentrations: there is NO new information related to SPAD values in comparison to the chlorophyll data!; why do you measure all these parameters?)

M&M section

Title for Chapter 2.4 not clear (you do not measure photosynthesis, but you measure chlorophyll concentrations)

Why do you measure SPAD values (see comments above)

Add references for the formulas in line 118-121

Why do you measure all these parameters ? most of the methods are not well escribed. You should add some scientific references (as many other papers are available)

L 145-151: obviously this section is a “copy&paste” section (soybeans??).

L155: move to section 2.2 & 2.3

Results and Discussion

Chapter 3.1. To my understanding, this is not a result, but these details should be given in the M&M section!

Overall the Results section is much too long (sse comments above: are all the parameters relevant? You might just mention that you select 1 “stress parameter”…)

Delete Figure 1 (no relevance for your interpretation!).

Overall, the use of colours in the figures is not clear. Treatments should get the same colour in all figures. Also, you should consider to indicate a kind of grouping with the colour (e.g. one colour for nano treatments and increasing colour tone with increasing Si concentration).

Figure 2: the picture with the petri dishes is not relevant

Unfortunately the effects of the increasing Si concentrations on shoot/root length are very “inconsistent”!!! Any explanation??

Figure 3: Delete the 2 shoot/root pictures

For several statements there is no backup by data from the experiment (e.g. L 233-234)

Chapter 3.5: you should always start the sub-chapters in 3.5 with results from your experiment and then discuss these results in the context of the literature.

Comments on the “antioxidant” system are rather general (L240-251).

It remains unclear why you mention results from an Fe NM (L266-268).

Chapter 3.6: it remains unclear why you have chosen those elements and do not show data on N/P/K/S/Ca! A lot of rather speculative statements (e.g. L297-303, 321-232)

Figure 6: Why do you use a different “red to blue” scale for shots/roots?

Overall there is a clear ned to use relevant references in the discussion parts of this chapter.

Conclusion

L361-370: this is just a kind of “summary” and not at all a conclusion!

References

9 out of 36 references are from the author group! (see as a touch of “self-citation”).

Supplement

Figure S1: Not at all relevant!

In general, the text is readable. Even though I am not a native speaker (and interestingly one of the co-authors is a native speaking person), I have the impression that not all parts of the text are written in an acceptable way. To my impression, the text needs considerable rewording and smoothing (e.g. line 38 “to narrow the gap between the existing yield”; L45 “as it can provide the release of trace elements or directly into the plant to promote plant growth”; L45-46; 50-51 “ … but most of these forms are not available to plants, which can only be absorbed in the monosilicate”;….).

I am sorry, but based on these arguments I recommend that your paper in its present form should not be accepted for publication in Nanomaterials, but needs a major revision.

Author Response

Dear reviewer,
We appreciate your valuable comments to help improve and strengthen our manuscript. We have now processed each of your comments and provided detailed responses below. Please refer to the attachment.

Reviewer 2 Report

The article describes the effect of silica-based nanomaterials on seed germination and seedling growth of rice.

The theme is interesting, although some concerns must be addressed:

-          Please improve the state-of-the-art, at this moment it looks like a technical report, not like a scientific one.

-          Try to highlight a little more the aim and novelty of the study. 

-          Please provide full information about used reagents and instruments (company, city, country).

-          Please replace H2O2 with H2O2 (use subscript)– line 78. Please verify all reagents in order to use subscripts.

-          Please use space between values and measurement units: for example, 27 °C, instead of 27°C as it was written in the manuscript.

-          Please number the equations.

-          Please define the terms from the equations.

-          Please define SOD, CAT and MDA.

-          All figures must be included after their first mention in the text.

-          Please provide more characterization details about the silica materials (specific surface area, FTIR, SEM).

-          Please specify whether or not the used materials were more efficient compared with other studies.

-          It is mandatory that the English language is revised by a native English speaker.

Author Response

(The authors gave the same response as above.)

Reviewer 3 Report

This is a well written paper on the effect of silica-based nanomaterials on rice seed germination and seedling growth.

I'd like to suggest the Authors to add more information about the specific rice variety that was selected for this study. Why the study was conducted on this specific hybrid? Genotype can play a big role in plant reaction and adaption to stresses and different varieties may have reached differently.

Similarly, the Authors should discuss why they chose those specific concentrations at which the silica-based NMs were applied.

There has been a lot of interest in NPs in the past 10 years and so many papers have been published on this topic. Adding more details will make your paper unique and will result in better reproducibility of the study.

Author Response

(The authors gave the same response as above.)

Round 2

Reviewer 2 Report

The manuscript can be accepted for publication in the present form.

Author Response

Dear reviewer,
We appreciate your valuable comments to help improve and strengthen our manuscript. We have now processed each of your comments and provided detailed responses below. Thank you again for your review.

Reviewer 3 Report

The Authors addressed all my comments and the manuscript is now improved and in a better shape.

However, there are some formatting issue that may need to be addressed. For examples in the introduction there are paragraphs numbered as 2.1 and 2.2. I am not sure if that is the correct format, since the introduction is considered chapter 1 and usually sub-paragraphs are not included.

Author Response

(The authors gave the same response as above.)
